# Non-Coding RNAs as Key Regulators in Lung Cancer

**DOI:** 10.3390/ijms25010560

**Published:** 2023-12-31

**Authors:** Irina Gilyazova, Galiya Gimalova, Aigul Nizamova, Elmira Galimova, Ekaterina Ishbulatova, Valentin Pavlov, Elza Khusnutdinova

**Affiliations:** 1Institute of Biochemistry and Genetics, Ufa Federal Research Center of Russian Academy of Sciences, 450054 Ufa, Russia; 2Institute of Urology and Clinical Oncology, Department of Medical Genetics and Fundamental Medicine, Bashkir State Medical University, 450008 Ufa, Russia; 3Department of Pathological Physiology, Bashkir State Medical University, 450008 Ufa, Russia; 4Institute of Urology and Clinical Oncology, Department of Urology, Bashkir State Medical University, 450008 Ufa, Russia

**Keywords:** lung cancer, non-coding RNA, miRNA, lncRNA, circRNA

## Abstract

For several decades, most lung cancer investigations have focused on the search for mutations in candidate genes; however, in the last decade, due to the fact that most of the human genome is occupied by sequences that do not code for proteins, much attention has been paid to non-coding RNAs (ncRNAs) that perform regulatory functions. In this review, we principally focused on recent studies of the function, regulatory mechanisms, and therapeutic potential of ncRNAs including microRNA (miRNA), long ncRNA (lncRNA), and circular RNA (circRNA) in different types of lung cancer.

## 1. Introduction

Lung cancer (LC) is the most frequently occurring cancer and the leading cause of cancer death in men. There were an estimated 2.2 million new cases and 1.8 million LC deaths worldwide in 2020 [1]. There are two main histological types of LC: non-small cell (NSCLC) and small-cell (SCLC) lung cancer, representing 80–85% and 15–20% of cases, respectively. NSCLC combines several forms of morphologically similar types of cancer with similar cell structures: squamous cell carcinoma (SCC), lung adenocarcinoma (LUAD), and large-cell carcinoma [2]. The disease develops latently for a long time. The tumour begins to form in the glands and mucosa; however, very quickly, metastases spread throughout the body. There are different risk factors for the disease including tobacco smoking, air pollution, viruses, and genetic factors as well.

Studies on the biology and genetics of LC have aided the realisation that this disease requires different approaches to treatment depending on the tumour features. It has been revealed that driver mutations are found in almost 50% of all the tumours in LC. The most important and frequent driver mutations in clinical practice are mutations in *EGFR* and the translocation of the *ALK* gene. Moreover, the roles of mutations in *ROS1*, *RET*, *BRAF*, *HER2*, *MET*, *RB1*, and *TP53* in LC development are also well known [3,4].

Cancer cells divide rapidly, spreading the tumour throughout the body, meaning timely diagnosis of the disease and optimal treatment are extremely important in increasing the chance of prolonging a patient’s life.

However, in addition to genetic changes, epigenetic changes can lead to the development of LC. Despite the fact that non-coding RNAs (ncRNAs) are regions of the transcriptome that do not code for proteins, they make up about 98% of the genome and their role is very important—they have a regulatory role in a huge number of biological processes, affect the expression of protein-coding genes, and can contribute to the development of various diseases, including cancer [5,6]. ncRNAs include long non-coding RNAs (lncRNAs), microRNAs (miRNAs), small interfering RNAs (siRNAs), piwi-interacting RNAs (piRNAs), promoter-associated transcripts (PATs), enhancer RNAs (eRNAs), and circular RNAs (circRNAs) [5].

The ncRNA family is classified into various subgroups determined by both the size and structure of the molecules. The most studied classes of ncRNAs are miRNAs, lncRNAs, and circRNAs, the functions of which are summarised in Figure 1.

ncRNAs with over 200 nucleotides are known as lncRNAs, whereas “small ncRNA” is a general term that encompasses all molecules shorter than 200 nucleotides [7]. miRNAs represent the most studied class among small ncRNAs. Originating from extended stem–loop structures, miRNAs possess the ability to bind to and suppress mRNAs [8]. The biogenesis of miRNAs involves a sequential process, beginning with their transcription as primary miRNAs (pri-miRNAs). Subsequently, in the nucleus, Drosha and DGCR8 undertake the processing of pri-miRNAs, leading to the formation of precursor miRNAs (pre-miRNAs). Once transported to the cytoplasm, pre-miRNAs undergo Dicer-mediated cleavage, resulting in the creation of an miRNA/miRNA duplex (Figure 2). Notably, only one of these miRNAs executes its inhibitory role, while the other undergoes degradation [9]. Furthermore, miRNAs exhibit atypical functions, including the inhibition of mitochondrial transcripts, activation of mRNA translation, binding and inhibition of proteins, coding for peptides, triggering of transcription, activation of Toll-like receptors, and inhibition of nuclear ncRNAs [10,11]. These varied functions underscore the complexity and versatility inherent in miRNAs as molecular entities.

miRNAs perform post-transcription regulation of mRNA expression and thus play a crucial role in the regulation of cell proliferation and differentiation, tumour metastasis, chemoresistance, and the epithelial–mesenchymal transition (EMT). For instance, some miRNAs are overexpressed during carcinogenesis (miR-17-92, miR-106a-363, miR-183/96/182, miR-34, etc.). Such miRNAs can be markers of resistance to therapy; on the other hand, regulation of their expression can affect patient outcomes.

miRNAs play their role in multidrug resistance development possibly via interaction with tumor-suppressing genes or oncogenes or by affecting the genes of proteins involved in drug transport. Another mechanism of drug resistance is the EMT, and some miRNAs (miR-495) can interact with factors linked to this process. Moreover, miRNAs can affect the expression of genes involved in DNA repair (miR-7-5p, miR-335) and thus change the response to therapeutic drugs causing DNA damage.

The clinical significance of miRNAs in cancer is also related to their diagnostic and prognostic roles. They are relevant for early disease diagnosis with specification of the tumour’s origin, its histological classification, and stage. Increased or decreased blood levels of some miRNAs can be of considerable interest in cancer diagnosis (miR-92, miR-34a, miR-21), and there is an association of miRNAs with tumour malignancy that is important in patient survival assessments.

Among ncRNAs, lncRNAs stand out as the most intricate. Serving as master regulators of gene expression within cells, lncRNAs are categorised into four classes — intronic, intergenic, antisense, and overlapping—based on their genomic positions [12]. lncRNAs undergo a biogenesis process akin to that of mRNAs, with many of them undergoing splicing, capping, and polyadenylation. The intricate 3D structure of these transcripts, characterised by rapid changes, is the source of their complexity, granting them the capacity to fulfill multiple functions [13,14]. lncRNAs exhibit both cis functions, which occur in the vicinity of their transcription site, and trans functions, which take place at a distance from the transcription site [15]. Common cis functions involve activities related to chromatin modifications, DNA transcription, and chromosomal looping. Distinct trans functions encompass binding to mRNAs, influencing their stability, interacting with proteins to modify their function, coding for micropeptides, engaging with other ncRNAs, and aiding in the assembly of paraspeckles [11,16,17].

lncRNAs are involved in processes of gene expression regulation, be it transcription and post-transcriptional processing of mRNA or DNA demethylation through different mechanisms. Specifically, lncRNAs promote cell proliferation and tumour progression, invasion, and migration, and can inhibit apoptosis and stimulate the EMT and metastasis. lncRNAs can also promote tumour development through interaction with miRNAs, including being a molecular sponge for them. In the context of clinical relevance, the effects of lncRNAs are realised in different ways. First, lncRNAs can serve as diagnostic markers, they can be used for tumour type differentiation and tumour staging (LUCAT1, MALAT1), and their expression can be related to tumour size or be an early marker of its development (MALAT1). Moreover, lncRNAs have prognostic value as markers of metastasis and as patient survival predictors. Thus, MALAT1, LUCAT1, and TUG1 are associated with the survival of patients with LC; HOTAIR can serve as a biomarker of tumour progression; and NEAT1 is related to poor prognosis. Additionally, lncRNAs are of considerable interest as treatment targets due to their effects on the EMT and resistant cell proliferation (UCA1, LINC00665, MEG3, and PVT1).

circRNAs, constituting the third major class of ncRNAs, are defined by their distinct structure. They form closed, uninterrupted loops where the 3′ and 5′ ends are covalently linked [18]. Notably, circRNAs are molecules characterised by tissue specificity and remarkable stability [7]. The formation of circRNAs is intricately controlled by both cis and trans-acting regulatory elements that modulate the splicing process [19]. There are several different mechanisms of circRNA biogenesis, with back-splicing identified as a universal event. Back-splicing can be initiated through the sequence complementarity of flanking introns, protein dimerization, intron lariat debranching, and exon skipping mechanisms [20]. Although the functions of circRNAs are only partially understood, these ncRNAs have been characterised as super-sponges capable of binding numerous miRNA molecules and impeding their function (Figure 3). Other well-known functions of circRNA include encoding of micropeptides [21], binding of proteins [22] with the ability to regulate their functions, and translation control [18,23,24].

circRNAs are implicated in gene regulation via the inhibition of miRNA activity. circRNAs are of interest in accurately diagnosing LC with a distinct epigenetic profile differentiating it from other lung pathologies. Moreover, they may serve as a promising marker for LC early diagnosis (circRNA-002178). Some circRNAs are related to metastasis and thus can be used as markers of this process (circSATB2, circRNA_102231). Other ones are studied as potential targets for treatment (circPVT1). It is also known that circRNAs associated with the survival of patients are correlated with lymphatic metastasis, the TNM stage, and chemotherapy resistance (circCRIM1, circPVT1, circTUBGCP3).

ncRNAs have great potential as diagnostic and prognostic biomarkers that can predict disease progression and response to therapy. Despite a fairly large number of studies on ncRNA in LC, research results remain conflicting. These studies aim to identify and explain the complex involvement of ncRNA in the pathogenesis of this disease. A comprehensive understanding of these mechanisms will not only facilitate the development of new biomarkers but will also open up new therapeutic targets.

This review systematises data on the role of the most investigated types of ncRNA in the pathogenesis of various types of LC.

## 2. miRNAs in LC

miRNAs are small endogenous RNAs with a size of 18–22 nucleotides that perform the function of recognizing complementary target sites in the 3’-untranslated region and subsequent suppression of mRNA expression in the post-transcription stage [25,26]. miRNAs are secreted into the extracellular space as components of exosomes or as complexes with proteins. These nucleic acids are involved in oncogenesis, metastasis, and the development of chemoresistance in tumour cells. miRNAs have been shown to play a crucial role in the regulation of cell proliferation and differentiation, and their expression is impaired in various diseases, especially in tumour progression [27]. In oncogenesis, miRNAs can perform two polar functions: on the one hand, they can act as suppressors of oncogenes; on the other hand, they can function as oncogenes themselves and inactivate tumour suppressors, stimulate tumour neoangiogenesis, and mediate immunosuppressive processes in neoplasms. This section discusses in detail the role of miRNAs in proliferation, metastasis, chemoresistance, resistance to targeted therapy and immunotherapy, regulation of apoptosis, autophagy, DNA repair, and the EMT. Examples of the use of miRNAs as diagnostic and prognostic biomarkers are also presented.

### 2.1. miRNAs in the Modulation of Cell Proliferation and Metastasis

Numerous studies have shown the involvement of miRNAs in the regulation of cell proliferation [28,29,30]. One of them revealed increased expression of 16 miRNAs and insufficient expression of eight miRNAs in SCLC cell lines and in primary SCLC tumour cells compared to normal lung tissue [30]. Among these 16 overexpressed miRNAs there were those that comprised parts of clusters, such as miR-17-92, miR-106a-363, miR-106b-25, miR-183/96/182. It was noted that miRNAs organised into one cluster showed increased or decreased expression together [30]. It is noteworthy that miRNAs can restrain the growth and dissemination of SCLC. miR-1 refers to miRNAs that have the property of suppressing tumour growth [30]. The *CXCR4* gene is a direct target of miR-1 in SCLC [29]. The involvement of this gene in the migration and adhesion of cancer cells, metastasis, and the development of chemoresistance has been noted [31,32]. It was reported that the introduction of miR-485-5p mimetics through transfection led to a discernible decline in the quantity of tumor cells, while the effect of inhibitors was the opposite [28]. In addition, the increased expression of miR-485-5p altered the invasion of tumour cells by targeting the Flotillin-2 (FLOT2) protein, which led to a decrease in the ability of tumour cells to penetrate into surrounding tissues [28]. Another miRNA, miR-886-3p, also acts as a tumour suppressor in SCLC. miR-886-3p is reported to modulate the proliferation of SCLC cells using Polo-like kinase 1 (PLK1) and transforming growth factor-beta 1 (TGF-β1) in vitro [33]. This miRNA inhibits the growth and invasion of SCLC tumours both in vitro and in vivo [34]. Cell proliferation is suppressed by miR-34b-3p and miR-27a-3p, while miR-485-3p additionally inhibits metastasis in SCLC [35]. Methylation of miR-34b/c has been observed to transpire in roughly 67% of SCLC tissues, resulting in the suppression of SCLC cell growth and metastatic potential [36]. The miRNA-34 family, a cluster of miRNAs associated with tumor suppression, is usually involved in the p53 signaling network [37]. miR-451a is insufficiently expressed in SCLC tissues, and it suppresses cell proliferation by suppressing lymphoid-specific helicase (HELLS) and regulates the expression of mTOR, a target of rapamycin in mammals, as well as the signaling pathways of apoptosis [38].

It has been reported that the increased regulation of miR-665 in malignant pleural effusion in patients with SCLC increases the cancer’s ability to invade, while its effect on cell growth remains unclear. Moreover, miR-665 participates in the regulation of HEY-like protein (HEYL), a target gene in the Notch signaling pathway. It has been observed that miR-665 suppresses the expression of epithelial markers but enhances the expression of mesenchymal markers, which indicates the crucial role of miR-665 in the EMT [39]. A study conducted on SCLC cell lines showed that tumour growth and invasion are suppressed by miR-450. miR-450 directly targets interferon regulation factor 2 (IRF2), which plays an essential role in tumour progression [40].

The role of miRNAs in SCLC carcinogenesis cannot be underestimated. SCLC tumour growth is greatly influenced by the overexpression of miR-134. By introducing miR-134 mimetics, the apoptosis of tumour cells is hindered, resulting in an increase in the level of anti-apoptotic genes and a decrease in the level of pro-apoptotic genes. The growth of cells in the H69 cell line is regulated by miR-134, which primarily targets the tumour suppressor gene oxidoreductase containing the WW domain (*WWOX*). This regulation occurs through the pathway of extracellular regulated protein kinases (ERK). The inhibition of miR-134 results in the opposite effect, whereas miR-134 mimetics hinder the phosphorylation of ERK [41]. SCLC cells, unlike normal lung cells, display an excessive expression of miR-543. Suppression of miR-543 leads to the hindering of both tumour cell proliferation and migration [42]. Apoptosis-related proteins, such as Bax/Bcl-2, caspase-3, and caspase-8, were found to be overexpressed, thereby enhancing the impact of apoptosis [42]. Fused in sarcoma 1 (*FUS1*), a tumour suppressor gene, is inhibited due to the overexpression of miR-93, miR-98, and miR-197. The overexpression of the three miRNAs in SCLC suggests their potential involvement in tumour progression [43]. In SCLC, miR-375 is highly expressed and regulates the expression of *ASCL1*, homologue one of the achaete–scute complex. ASCL1 plays a role in the development of neurons and neuroendocrine cells and activates the expression of miR-375 [44]. The targeting of inositol 1,4,5-triphosphate-3-kinase B (ITPKB) leads to a significant increase in miR-375 expression in SCLC, resulting in enhanced cell growth and reduced ITPKB protein levels [45].

miRNAs influence cancer cell cycles. For instance, miR-126 suppresses cancer cell growth, stopping it in the G1 phase of the cell cycle by targeting protein 5 of solute carrier family 7 (SLC7A5) [46]. Additionally, the expression of the *TSPAN12* gene, which encodes the tetraspanin-12 protein, is regulated by miR-495, leading to the inhibition of SCLC tumour growth [47]. Unlike control cell lines, miR-216a-5p expression was found to be suppressed in SCLC cell lines [48]. Further study results demonstrate that miR-216a-5p activation can lead to the suppression of tumour graft progression and migration in mouse models. Cell cycle disorders occur as a result of the suppression of miR-216a-5p expression, which leads to SCLC cells being stopped in the G2/M phase. miR-216a-5p also can negatively regulate the expression of Bcl-2 family proteins [48].

An unfavorable prognosis for cancer patients is often associated with tumour angiogenesis, as it promotes tumour growth, proliferation, and metastasis. Therefore, it is crucial to gain a comprehensive understanding of the molecular mechanisms underlying this process. It has been found that the levels of miR-141 are notably elevated in plasma exosomes and serum samples taken from individuals with SCLC [49]. Furthermore, the presence of high levels of miR-141 is associated with advanced stages of the disease as classified by TNM, as well as an increased invasiveness of tumours. Possible treatment strategies can be indicated by inhibiting the growth of SCLC by targeting miR-141 and Kruppel-like factor 12 (KLF12), which would inhibit angiogenesis. miR-141 affects angiogenesis through the regulation of *KLF12* expression. miR-141 overexpression promotes SCLC tumour growth, migration, and angiogenesis in vivo [49].

The role of miRNA in tumour metastasis is of great importance. It has been found that miR-184 inhibits SCLC metastasis, while miR-574-5p has the opposite effect. According to bioinformatic analysis, both miR-184 and miR-574-5p target the proteins endothelial PAS domain protein 1 (EPAS1) and protein tyrosine phosphatase receptor type U (PTPRU). By regulating β-catenin signaling, these miRNAs suppress the expression of *PTPRU* or *EPAS1* [50]. The presence of lymph node (LN) metastasis is considered a clinical indicator of tumour stage, yet the relationship between miRNA and LN metastasis remains poorly investigated. The involvement of miR-126 in the metastasis of SCLC tumour to LNs was investigated in a study utilising the Gene Expression Omnibus (GEO) database [51]. According to the TNM classification, higher expression of exosomal miR-375-3p is observed in patients with later stages of SCLC [52].

The destructive effects of exosomal miR-375-3p on vascular barriers result in increased permeability, ultimately promoting cancer migration and metastasis [52]. The mechanism of metastasis of tumour cells into the blood is indicated by a similar phenomenon occurring for the structural barriers of lung, liver, and brain tissue cells [52].

The enhancement of tumour cell proliferation in NSCLC has been attributed to the discovery of several miRNAs. Specifically, miR-150-5p and miR-106a-5p have been shown to suppress LKB1 and promote NSCLC cell proliferation [7,53,54]. The research conducted by Chang et al. [55] revealed that LUAD cells secreted extracellular vesicles containing miR-150-5p. These vesicles have a significant impact on the tumour microenvironment by inducing immunosuppressive conditions and reducing *CD226* expression on NK cells. As CD226 is involved in tumour growth control, the inhibition of miR-150-5p leads to the activation of cytotoxic NK cells through the functioning of CD226 [55].

The potential of miRNAs to improve treatment outcomes (miR-200c-3p [56]) by reducing resistance or contribute to resistance development is well-documented. Various studies have demonstrated tumour-suppressive effects of miR-218-5p, miR-370, miR-377, and miR-646 by targeting *EGFR* in multiple NSCLC cell lines [57,58,59,60]. In vivo experiments have also shown that miR-370 and miR-218-5p exhibit antitumour activity [58,60]. Additionally, miR-183-5p, miR-142-5p, and miR-1261 have been identified as tumour suppressors in NSCLC by targeting *PIK3CA* [61,62].

The low immunogenicity and toxicity of liposomes have garnered much attention in recent times. These biocompatible spherical bubbles are composed of lipids naturally found in cell membranes. Lipoplexes, which are structures formed when RNA molecules combine with lipids, have also been extensively studied [63]. Jiang et al. conducted an in vivo study [64] which demonstrated that the delivery of miR-143 through lipoplexes effectively inhibits the metastasis of NSCLC.

### 2.2. miRNAs in the Modulation of Chemoresistance

The standard treatments for LC include surgery, radiation therapy, chemotherapy, and immune checkpoint inhibitors. However, the limited effectiveness of these treatments is attributed to chemoresistance, which can result from various mechanisms. These mechanisms encompass abnormal signaling, impaired regulation of DNA damage repair, suppression of autophagy and apoptosis, and impaired expression of transport proteins [25]. miRNAs have been found to play a vital role in the mechanism of multidrug resistance in SCLC cell models. A large-scale study was conducted to evaluate the expression of 856 miRNAs in these cell models using a microchip. The interaction between the mechanisms involved in multidrug resistance is not fully understood, but this study shed some light on the role of miRNAs. In the study, 61 miRNAs were found to be differentially expressed in cell lines that were either sensitive or resistant to chemotherapy. Among these miRNAs, 24 showed an increase in expression while 37 showed a decrease in expression [65]. In another study, it was found that the expression of miR-20a in exosomes obtained from tumour-associated fibroblasts of patients with NSCLC is increased, and this contributes to the emergence of resistance of NSCLC cells to cisplatin [66]. Phosphatase and tensin homologue (*PTEN*), the overexpression of which suppresses NSCLC cell proliferation and chemoresistance, is the target gene for miR-20a [66]. The regulation of drug transport is important in the formation of chemoresistance. Multidrug resistance protein 1 (MRP1) belongs to the superfamily of ATP-binding cassette (ABC) transporters and is encoded by the *ABCC1* gene [67]. MRP1 carries various endogenous and exogenous substances, including anticancer drugs. High expression of MRP1/*ABCC1* in cells provides resistance to various antitumour drugs, such as vincristine, etoposide, mitoxantrone, and doxorubicin [68]. A study by Liu et al. allowed us to establish that MRP1/*ABCC1* is the target of miR-7 in SCLC [69]. A number of studies have described miRNAs that affect the development of resistance to the antitumour drug cisplatin in NSCLC [70,71,72]. Hao and colleagues concluded that cisplatin resistance in NSCLC is due to overexpression of miR-369-3p, which directly regulates the expression of the *SLC35F5* gene [70]. This gene is a transporter of nucleotide sugar and participates in the process of drug assimilation [70]. According to a different study, the development of cisplatin resistance in NSCLC cells is associated with the diminished expression of miR-185-5p, which inhibits the expression of the *ABCC1* gene. This study specifically examined human pulmonary adenocarcinoma cell lines that were either sensitive or resistant to cisplatin [71]. Interestingly, the expression of ATP7A, which is a copper transporter and regulates the excretion of cisplatin from cells, is regulated by miR-495-3p and affects resistance to cisplatin in NSCLC [72].

The chemoresistance that occurs during the treatment of patients with LC significantly complicates the fight against this disease. Understanding the molecular basis of this phenomenon can become an information base necessary for monitoring and predicting drug resistance.

### 2.3. miRNAs in Resistance to Targeted Therapy and Immunotherapy

Patients with NSCLC harbouring epidermal growth factor receptor (EGFR)-activated mutations are prevalent in clinical practice and typically experience positive outcomes with EGFR tyrosine kinase inhibitors (TKIs). The first-generation EGFR-TKIs, such as gefitinib and erlotinib, play a crucial role in the management of LC. However, it is imperative to acknowledge the emergence of resistance to these agents [73]. Numerous studies have demonstrated the involvement of miRNAs in the development of this resistance. One study showed that reduced miR-506-3p expression contributed to the development of erlotinib resistance in an NSCLC cell line through activation of the Sonic Hedgehog signaling pathway [74]. Another study revealed the oncogenic potential of miR-135a. Overexpression of miR-135a conferred resistance to gefitinib in NSCLC cells by regulating the activity of *RAC1* and the PI3K/AKT signaling pathway [75]. TKI resistance in LC may be due to reciprocal activation of the YAP1/ERK axis mediated by downregulation of miR-630 [76]. The shedding of exosome-transmitted miR-522-3p by NSCLC cells resistant to EGFR-TKI with the *T790M* mutation has the ability to induce gefitinib resistance in sensitive cells. This induction occurs through the activation of the PI3K/AKT signaling pathway, as shown by Liu et al. [77]. In a study conducted by Cao et al., it was found that the reduced expression of miR-19a in NSCLC cells contributed to the emergence of resistance to gefitinib and the EMT by regulating the expression of c-Met, AKT, and ERK signaling pathways [78].

Studies conducted by researchers from various countries have duly ascertained the crucial significance of miRNAs in the development of resistance against gefitinib and erlotinib, thereby positing that targeting these specific miRNAs could potentially serve as a highly efficacious therapeutic approach for patients afflicted with first-generation EGFR-TKI-resistant LC.

Immunotherapy, an increasingly prominent modality, has achieved remarkable success in revolutionising cancer treatment. Significantly, LC has not been exempted from benefiting from this advancement. In recent times, multiple well-designed and rigorous randomised controlled trials have notably substantiated that immune checkpoint inhibitors (ICIs), when employed, can substantially enhance the overall survival (OS) rates for patients confronting advanced LC, surpassing the efficacy of traditional chemotherapy approaches [79,80,81,82]. In particular, immunotherapy employing anti-PD-1 or anti-PD-L1 antibodies has emerged as a prominent therapeutic approach for individuals afflicted with advanced NSCLC and SCLC [83,84,85,86]. In spite of notable clinical advancements, a majority of patients experience suboptimal responses to ICI therapy due to the emergence of primary or secondary resistance [73].

The relative expression of miRNAs in patients afflicted with NSCLC who were subjected to immunotherapy using pembrolizumab or nivolumab was examined by Grenda et al. Accordingly, significant upregulation of miR-200b and significant downregulation of miR-429 were discerned in individuals who demonstrated positive responses to immunotherapy in contrast to patients who did not exhibit any notable response [86]. A group of researchers from France compared miRNA expression profiles in the plasma of patients with advanced NSCLC before and during treatment with nivolumab. Patients who benefited from treatment had decreased expression of both miR-320b and miR-375, which significantly distinguished them from patients with early progression [87]. A research investigation conducted on a Chinese cohort revealed the presence of 27 serum miRNAs that exhibited differential expression in patients with advanced NSCLC who demonstrated positive responses to nivolumab therapy in contrast to those who did not respond. Of these 27 serum miRNAs, 22 displayed a significant increase in expression levels among those who responded to the therapy, whereas five miRNAs exhibited a significant decrease in expression levels compared to the non-responders [88].

Studying the role of miRNAs in regulating the response of patients with LC to immunotherapy is of great interest. New knowledge about this problem can significantly influence the development of personalised effective immunotherapy.

### 2.4. miRNAs Regulating Apoptosis and Autophagy

The modulation of apoptosis by miRNAs plays a crucial role in regulating the responsiveness of SCLC cells to chemotherapy. Specifically, miR-494 targets secretagogin (SCGN), which is involved in the apoptosis processes of tumour cells. Overexpression of *SCGN* leads to an increase in chemosensitivity by suppressing apoptosis through the regulatory influence of miR-494 [89]. Additionally, previous studies have highlighted the abnormal expression of Hox in various cancer tissues, such as prostate cancer and LC [90]. *HOXA1* overexpression promotes apoptosis and enhances the chemosensitivity of cancer cells, whereas the inhibition of *HOXA1* has the opposite effect [91]. In a luciferase reporter assay, it was observed that miR-100, when overexpressed, enhances cell resistance to chemotherapy and decreases the expression of *HOXA1* [91]. Moreover, *TSPAN12*, when highly expressed, promotes chemoresistance in SCLC cells through the regulation of miR-495. Conversely, downregulation of *TSPAN12* expression causes apoptosis to increase, leading to enhanced chemoresistance and cell cycle arrest in the G1 phase [47]. Y-box 2 (SOX2) inhibits drug-induced apoptosis in cells, thereby inducing resistance to cisplatin [92]. In cisplatin-resistant cancer cells, there is an observed overexpression of *SOX2*. Furthermore, there is an underexpression of miR-340-5p in these cells [92]. The signaling of the apoptotic pathway is regulated by Bcl-2, an oncogene [93]. The underexpression of miR-181b induces cisplatin resistance in SCLC cells. However, this resistance can be inhibited by a higher level of miR-181b, which targets Bcl-2 [94]. On the other hand, miR-339-5p is underexpressed in Taxol-resistant cells. In vitro studies have suggested that miR-339-5p promotes tumour growth and suppresses apoptosis in SCLC cells by targeting α1,2-fucosyltransferase 1 (*FUT1*) [95].

Autophagy, a process responsible for the degradation of damaged proteins and cellular components, is known to play an important role in the development of diseases and drug resistance [96]. The discovery of the first autophagy-related protein in mammals, Beclin-1, revealed its increased activity in SCLC cells resistant to etoposide/cisplatin. In order to enhance the effectiveness of chemotherapy, the expression of Beclin-1 can be reduced by overexpressing miR-30a-5p [97]. Additionally, the expression of miR-24-3p was found to be suppressed in etoposide/cisplatin-resistant SCLC cells. The suppression of miR-24-3p leads to the activation of autophagy through the targeting the *ATG4A* gene [98].

### 2.5. miRNAs Regulating DNA Repair and the EMT

In the field of cancer therapy, the DNA-damage agent has gained immense popularity as a therapeutic drug, effectively inhibiting cell division through the induction of double-strand breaks. Interestingly, tumour cells have developed a mechanism to counteract the effects of this agent by regulating the pathway responsible for DNA damage repair. Within this context, it has been observed that miR-7-5p is significantly downregulated in SCLC cells that display resistance to the DNA-damage agent doxorubicin. This miRNA specifically targets a protein called poly-ADP-ribose polymerase 1 (PARP1) [99,100]. The downregulation of *PARP1* expression by miR-7-5p has been found to exert suppressive effects on the process of homologous recombination repair [99]. Additionally, miR-335 has also been identified as a regulator of *PARP1* expression, displaying an inverse association with the levels of PARP-1 expression. In resistant SCLC cells, miR-335 exhibits lower expression levels, potentially due to its targeting of PARP-1 [101]. It is therefore evident that miR-335 plays a significant role in mediating the chemosensitivity of SCLC cells.

One of the mechanisms of drug resistance is believed to be the EMT, whereby cells undergo a change in phenotype from epithelial to mesenchymal. The transition in question is facilitated through the involvement of multiple factors, notably vimentin and the zinc-finger E-box-binding (ZEB) protein. The EMT has been shown to promote the invasion and metastasis of tumour cells [101]. The connection between the EMT and drug resistance is increasingly being studied, with breast cancer being one example where this link has been observed [102]. Epithelial and endothelial tyrosine kinase (Etk/*BMX*) has emerged as a pertinent target gene of miR-495, an miRNA known for its involvement in drug resistance within SCLC by virtue of repressing apoptotic processes.

Furthermore, the overexpression of Etk/*BMX* enhances the regulation of the EMT mesenchymal molecules such as Twist, Vim, and ZEB [103]. Additionally, ZEB2 involvement in the EMT is observed through the NF-kB, Notch, and TGF-β signaling pathways. An analysis of SCLC tumour tissue showed that *ZEB2* suppression and miR-200b overexpression result in increased sensitivity to chemotherapeutic drugs. Moreover, miR-200b mediates the expression of *ZEB2* by binding to *ZEB2* 3′UTR [104].

### 2.6. Clinical Use of miRNAs as Diagnostic and Prognostic Biomarkers

Due to the limited availability of effective therapeutic options in LC and low survival rates after treatment, the early detection of LC is of utmost importance for improving the prognosis. Traditionally, the histological diagnosis heavily relied on biopsy, which enabled the examination of cell morphology, and immunohistochemistry, an invasive investigative method. miRNAs, as new diagnostic biomarkers, offer the potential of noninvasive detection, making them suitable for large-scale LC screening for early disease diagnosis [105].

One study observed a significant distinction in the expression profile of miRNAs between SCLC cells and normal tissue cells. This was confirmed through bioinformatic analysis, which identified 56 and 135 differentially expressed miRNAs in SCLC and normal tissue cells, respectively. These findings support the role of miRNAs in distinguishing patients with SCLC from healthy individuals [106,107]. In addition, a separate investigation examined the diagnostic potential of miR-92-a2 and discovered that plasma levels of miR-92a-2 were notably higher in SCLC patients compared to the control group. The study determined the sensitivity and specificity of miR-92a-2 as diagnostic markers for SCLC to be 56% and 100%, respectively, with an area under the receiver operating characteristic curve (AUC) of 0.761 [108]. These findings suggest that miR-92a-2 can be utilised as a diagnostic biomarker for SCLC.

Additionally, the differentiation of NSCLC from SCLC could be aided by miRNAs. A total of 31 SCLC tumours were analyzed to study the expression of seven miRNAs. The results showed that there was a lower number of miRNAs including miR-34a, miR-29b, and miR-21 in the cell lines of SCLC [109]. The expression profiles of miRNAs in SCLC and NSCLC were compared in another study, revealing the overexpression of 19 miRNAs and the decreased expression of 10 miRNAs [110]. To further investigate the ability of miRNAs to accurately diagnose SCLC and NSCLC, bronchial brushing samples were examined using quantitative RT-PCR. The combination of miR-29a and miR-375 yielded an accuracy of 86.5% (AUC 0.947) [111]. Additionally, the diagnostic efficacy of miR-375, miR-17, and miR-190b was found to be favorable. In the training cohort, the AUC reached 0.878, with specificity and sensitivity levels of 80% and 81%, respectively. Similarly, in the validation cohort, the AUC was recorded at 0.869, accompanied by specificity and sensitivity rates of 84% and 82%, respectively [112]. Furthermore, the miRview diagnostic test has been developed for the classification of LC subtypes, achieving an overall accuracy of up to 94%.

A recent study has shown that miRNA levels may be associated with tumour malignancy, as different SCLC tumours at the same stages of development exhibited varying survival rates and prognoses for patients [113]. Consequently, the use of miRNA as a diagnostic tool has the potential to provide more accurate determinations of SCLC. Nonetheless, it should be noted that miRNA has been found to be ineffective in distinguishing SCLC from benign lung disease [114].

In the context of previous discussions, the crucial role played by miRNAs in the regulation of acquired drug resistance, radiation therapy, immunotherapy, and targeted therapy has been emphasised. Consequently, several investigations have been conducted to explore the potential of these molecules in tracking or predicting drug resistance. In a notable study, the implementation of RNA sequencing was utilised to examine the SCLC tissues sourced from six patients who displayed either a partial response (PR) or stable disease (SD)/progressive disease (PD) subsequent to first-line chemotherapy [115]. The outcomes of this analysis revealed the existence of 1303 distinct miRNAs between the PR and PD or SD groups, characterised by 520 miRNAs exhibiting upregulation and 783 demonstrating downregulation in the PR group [115]. Furthermore, decreased expression levels of miR-596 and miR-601 were discerned in patients with PR when compared to those in the PD and SD groups, suggesting the potential utilization of miR-596 and miR-601 as biomarkers for anticipating chemotherapy resistance in SCLC patients [115]. Additionally, the upregulation of miR-92a-2* was found to be associated with chemoresistance, suggesting its utility in screening SCLC patients at risk of developing drug resistance [116]. Diagnostic significance was also demonstrated for miR-375 and miR-92b, as they exhibited activation following the onset of chemoresistance. According to the results of the ROC analysis, the AUC values for the two variables were found to be 0.766 and 0.791, respectively, as mentioned in [117].

miRNAs have undergone thorough research regarding their potential role as predictive biological indicators in SCLC. Through the application of Kaplan–Meyer survival analysis, it was determined that a decrease in progression-free survival (PFS) in SCLC patients is associated with miR-375 and miR-92b [117]. Interestingly, the expression levels of miR-375 and miR-92b show a significant decrease following effective chemotherapy [117]. Another study identified 11 miRNAs that are correlated with OS, with seven miRNAs showing a positive correlation with OS in SCLC patients, while the remaining four miRNAs demonstrated a negative association [118]. The random forest method was employed to conduct a more detailed analysis of hsa-miR-608, hsa-miR-9, and hsa-miR-194 [118]. By devising an equation that takes into account the complex Cox regression coefficient based on the expression levels of three specific miRNAs, it was possible to distinguish between patients with SCLC who had a longer OS and those who had a shorter OS [118]. The expression level of miR-92a-2*, on the other hand, was found to be inversely correlated with survival time in SCLC patients [116]. Furthermore, prognostic models were developed by studying OS-related miRNAs and subsequently validated in an independent cohort of 40 cases utilising quantitative RT-PCR [119]. The presence of both miR-886-3p and miR-150 in the biomarker profile was significantly correlated with OS in the initial and validation cohorts [119]. This indicates that miRNA profiles indicating a higher risk are associated with reduced OS and PFS in contrast to profiles denoting a lower risk [119]. Additionally, the findings continued to show statistical importance even when modifications were made for variables such as gender, age, and tobacco use. This indicates that the miR-150/miR-886-3p marker might act as a standalone predictor of disease progression in SCLC [119].

## 3. lncRNAs in LC

lncRNAs are transcripts composed of more than 200 nucleotides that do not serve as blueprints for protein synthesis. These lncRNAs constitute roughly 70% of all ncRNA types. The human genome is rich in lncRNAs, which are crucial for a wide range of biological functions (Figure 4). Most of these transcripts undergo polyadenylation and splicing and are found in the nucleus and, in particular, in chromatin-associated fractions [120,121]. According to their localization, many lncRNAs are associated with the regulation of gene expression and the formation of the three-dimensional organization of the nucleus. lncRNAs play important roles in different physiological and pathological processes in cells interacting with DNA, RNA (including miRNA), and various proteins [122]. It has been shown that lncRNAs can interact with some proteins involved in transcription, and they can also control various steps in the post-transcriptional processing of mRNA.

lncRNAs also play an important role in DNA imprinting and demethylation, RNA interference, chromatin remodeling, etc. [121].

Although some lncRNAs are found in introns, most of them are transcribed, capturing regions of sense and noncoding gene sequences [123].

Being involved in the key processes of regulation of gene expression, lncRNAs can act as oncogenes or tumour suppressors along with protein-coding genes (Table 1).

Mutations and/or epigenetic changes can lead to changes in the expression and structural and functional characteristics of one or another lncRNA, provoking neoplastic tissue growth. The oncogenic effect in each specific case will be due to the functional purpose of lncRNA in molecular cellular pathways.

Analysis of gene expression in tumour and normal cells revealed changes in lncRNA expression in several forms of cancer.

It has been shown that changes in lncRNA expression profiles due to alterations in epigenetic regulation or the influence of local active compounds in tumour microenvironments can be the first sign of LC development.

The expression study showed that many DNA regions that do not encode proteins are differently expressed in different stages of human cancer [146]. Analysis of chronic lymphocytic leukemia, colorectal cancer, and hepatocellular carcinoma revealed that all three types of cancer have similar lncRNA expression profiles compared to normal cells. Further analysis of one of the lncRNAs showed that it behaved like an oncogene, blocking apoptosis and leading to an increase in the number of malignant cells [146]. It is likely that these lncRNAs, which exhibit an abnormal level of expression during malignant transformation, perform important functions in the early stages of embryogenesis.

Any lncRNA functions according to its nucleotide sequence and secondary structure; they may form duplexes, loops etc. that can represent binding sites for diverse proteins. Some lncRNAs can form spatial structures that can serve as a base for the assembly and functioning of multicomponent protein complexes. In particular, in the processes of oncogenesis, the activity of protein systems formed by histone-modifying proteins and chromatin modifier proteins is often impaired.

### 3.1. lncRNAs as Tumour Promoters

One example of an lncRNA involved in carcinogenesis at this level is the HOTAIR (HOX antisense intergenic RNA) transcript, which is overexpressed in many cases of neoplastic tissue transformation. It has been shown that this lncRNA is involved in at least two mechanisms of epigenetic regulation of the expression of tumour suppressor genes. HOTAIR is a specific negative regulator of a number of so-called homeobox genes, directing proteins of the inhibitory complex 2 (PRC2) to the *HOXD* locus located on the second chromosome [124]. Homeobox genes play a key role in the processes of apoptosis, cell signaling, motility, and angiogenesis; therefore, disruption of their expression often has an oncogenic effect [147]. In addition, HOTAIR interacts with the histone-modifying complex, participating in histone methylation and demethylation, non-specifically inhibiting the expression of some tumour suppressor genes, such as *PTEN* and *GDF15* [125,126].

Another example of an lncRNA involved in carcinogenesis is MALAT1 (metastasis-associated lung adenocarcinoma transcript 1)—a component of spliceosomes. It has been shown that this lncRNA regulates the cellular level of active forms of SR proteins involved in alternative splicing [148]. Increased expression of MALAT1 causes increased cell proliferation by altering the level of pre-mRNA processing of transcription factors involved in the cell cycle [127]. Moreover, MALAT1 can inhibit cancer cells apoptosis and thus stimulate metastasis [127], being activated in early NSCLC, and overexpression of MALAT1 is an early prognostic marker for patients [138,139]. Interestingly, in cells, MALAT1 is localised in regions associated with the processing of pre-mRNA and so possibly can play a role in gene regulation. Moreover, studies have shown that MALAT1 can promote breast cancer while interacting with miRNA miR-1, and their expression is negatively correlated.

Similarly, PVT1 is also recurrently overexpressed in lung tumours, and changes in its level alter cell growth and invasiveness. Expression of this lncRNA correlates with expression of the *MYC* oncogene in different types of cancer. A decrease in PVT1 levels leads to reduced *MYC* levels. Therefore PVT1 is a promising target for the treatment of MYC-driven cancers via indirect influence on this transcription factor [128].

LUCAT1 (lung cancer-associated transcript 1) lncRNA is shown to be overexpressed in several cancer types including smoking-related LC [129]. The expression of LUCAT1 in LC samples is related to the tumour size, TNM stage, and OS. Intracellularly, LUCAT1 is located in the nucleus and cytoplasm. In the nucleus, LUCAT1 is involved in gene transcription regulation. It plays a role in the modification of tumor suppressor methylation through the activation and regulation of expression of the DNMT1 protein [149]. Reduced levels of this lncRNA influence the cell cycle, promoting cell apoptosis and inhibiting cell proliferation. LUCAT1 interacts with STAT1 and chromatin, thus showing an immune suppressive activity. Moreover, it can inhibit NF-kB functions [130]. The role of LUCAT1 located in the cytoplasm is probably related to the post-transcriptional and translational regulation of genes. Studies on pancreatic adenocarcinoma demonstrated that this lncRNA can promote cancer, acting as a molecular sponge for miR-539 as well [150]. Therefore, LUCAT1 can be considered a tumour-promoting lncRNA with complex intra- and extracellular activity.

Oncogenic activity has also been found for LINC00680, which acts via binding to *GATA6*, [131,132] and LINC00511, which binds *EZH2* and the chromatin modifying enzyme and represses p57, *LATS2*, and some other tumour-suppressor genes [133].

### 3.2. lncRNAs as Tumour Suppressors

A tumour-suppressive role in LC (NSCLC) was discovered for GAS5 (growth arrest-specific 5) lncRNA. GAS5 levels were found to be reduced in tumour specimens, which was associated with increased tumour dimensions, decreased differentiation, and more advanced stages of tumour-node metastasis [134]. It has been proposed that this lncRNA plays a role in cell cycle regulation through its influence on corresponding gene activities [151,152].

Research has indicated that GAS5 has the capability to engage miRNAs via base pairing and collaborate with various functional proteins. This interaction serves to impede their biological activities by affecting signaling pathways (as illustrated in Figure 5), as well as modifying the levels of autophagy within cells, oxidative stress, and the performance of immune cells in a living organism. Furthermore, GAS5 plays a role in controlling cell proliferation, invasiveness, and programmed cell death through the aforementioned molecular processes [153].

GAS5 notably suppresses the expression of miR-205 in NSCLC, with miR-205 known to engage with PTEN [154]. Further, Xue et al. [155] demonstrated that GAS5 substantially reduces miR-135b levels, thereby enhancing the responsiveness of NSCLC to radiotherapy.

A tumour-suppressive function has also been discovered for MEG3 and TUG1 [135].

It has been shown that the overexpression of MEG3 can induce apoptosis and regulates the cell cycle that arrests tumour cell growth [136]. In a study on NSCLC cells, MEG3 was found to affect the expression of p53, leading to suppression of the proliferation and induction of apoptosis [156]. Later, the inhibition of EMT via downregulation of MEG3 was also demonstrated [157].

The TUG1 (taurine-upregulated gene 1) lncRNA is overexpressed in many cancers, affecting cancer cell proliferation and apoptosis. It regulates gene expression by sponging specific miRNAs. Thus, in esophageal SCC, it stimulates cell proliferation and invasion via miR-498, which regulates cell division cycle 42 (*CDC42*) expression [158]. Moreover, the promoter region of TUG1 has a p53-binding site and is considered to be a direct target of p53 transcription. Loss of p53 expression could lead to the downregulation of TUG1 lncRNA in NSCLC [159].

### 3.3. lncRNAs as Diagnostic and Prognostic Biomarkers

A range of genomic research has shown that NSCLC is a diverse illness characterised by distinct molecular signatures that may have significant implications for clinical practice, and lncRNAs can help improve the prognoses of patients.

Some lncRNAs show diagnostic and predictive value in cancer treatment. For example, MALAT1 is associated with the survival of patients with LUAD and SCC. This is closely related to the histology and staging of metastasis in patients with NSCLC [160,161,162]. Moreover, MALAT1 has been shown to promote brain metastasis by inducing the EMT in LC [162].

HOTAIR lncRNA demonstrated relatively stable levels in plasma, meaning that it can be used as a biomarker of tumour progression in case of any changes in this level.

High levels of LUCAT1 in tumours are associated with poor OS. This may be due to repression of tumor suppressors’ p21 and p57 expression [163]. In addition, high LUCAT1 levels have been associated with late-stage tumour metastasis to the LNs and large tumour volumes [129].

Certain studies established SPRY4-IT1 expression as an independent risk factor for the prognosis of NSCLC, having an association with specific clinicopathological features [140].

lncRNA NEAT1 is also highly expressed in NSCLC tissues, and this is related to the poor prognosis of NSCLC patients [141].

TUG1 expression can also be a marker of LC patients’ prognoses [164].

### 3.4. lncRNAs as Biomarkers of Response to Treatment

Different studies show that lncRNAs may influence the reaction of NSCLC to cisplatin therapy through the control of regulators involved in the cell cycle and cell death. Cisplatin resistance in cancer cells arises through multiple pathways, and one such pathway involves the EMT process.

Inhibition of UCA1 lncRNA could invert the EMT process, thus leading to the restoration of the resistance of A549/DDP cells to cisplatin and improving the sensitivity to therapeutic drugs. Therefore, UCA1 expression is associated with the grade and stage of tumours and the resistance of tumour cells to chemotherapeutic agents [137].

Another lncRNA related to the cisplatin resistance is H19. Studies on LUAD showed elevated levels of this lncRNA in cisplatin-resistant cells, suggesting that it can mediate drug responses by regulating apoptosis and cell migration [142].

LUCAT1 expression is also found to be increased in NSCLC cells resistant to cisplatin [165]. This effect may develop because it targets IGF-2 involved in the regulation of cell proliferation and causes its overexpression.

The efficacy of EGFR-TKIs in NSCLC treatment can also be affected by some lncRNAs. LINC00665 can be considered to be among the predictors of this therapeutic agent’s efficacy. The suppression of this element may hinder the growth of cells resistant to gefitinib, enhance apoptosis, and decrease the movement of cells in a controlled environment, as well as impede the formation of tumours in cells with gefitinib resistance when studied within a living organism. Reducing LINC00665 levels could markedly diminish the compromised activation of EGFR and AKT [143].

BC087858 lncRNA is also related to EGFR-TKI resistance in NSCLC. Higher levels of BC087858 can activate the EMT and the PI3K/AKT and MEK/ERK pathways, and so may have significant part in overcoming non-*T790M*-acquired resistance [144].

Once more, the lncRNA MEG3 influences various mechanisms, notably triggering the mitochondrial apoptosis pathway via p53 activation and decreasing Bcl-xl levels, thereby modulating the sensitivity to cisplatin in NSCLC [145].

## 4. circRNAs in LC

circRNAs are non-coding RNAs that regulate the initiation and progression of various human diseases, including cancer. Their ends are circled with the help of a covalent bond between the terminal nucleotides. The ends of the 5’- and 3’-transcripts of circRNA are connected by a phosphodiester bond, resulting in the creation of a ring structure. The formation of circRNAs is facilitated by inverted repeats contained in their precursors. It is believed that the role of circRNA is to regulate gene expression by inhibiting miRNA activity. Since the discovery of the first circRNAs, they were considered to appear as a result of aberrant splicing and were not given any important fundamental significance [166]. However, with the rapid development of technologies for molecular biological, genetic, and bioinformatic analysis, their functions and roles in the development of complex diseases, including cancers, have become increasingly clear. Previously, several investigations have discovered that circRNAs are associated with different types of cancers [167,168]. circRNAs are involved in various processes realising the tumour potential of cells (Table 2).

It was found that circRNAs are involved in the formation of the tumour microenvironment. Numerous studies have shown that most of the known circRNAs function as pro-oncogenes and stimulate cell proliferation, angiogenesis, and tumour growth. circRNAs have multiple biological functions [191]; they are known to play the role of “sponges” for miRNAs and thus they regulate gene expression at the post-transcriptional level [192]. Features of circRNA such as stability and a presence in various tissues, including blood and other biological liquids, make them a possible candidate for the role of a diagnostic and prognostic biomarker that can aid in the early diagnosis of cancers and predict recurrence and metastatic disease. However, information on the underlying mechanisms and clinical significance of circRNAs in LC still remains scant and their role and functions need to be explored.

### 4.1. circRNAs as Diagnostic Biomarkers

circRNA analysis is a promising approach that can be used to differentiate benign lung changes from LC. By examining the patterns of circRNA analysis, we can identify distinct epigenetic signatures associated with LC.

A group of researchers from China studied the mechanism of LC progression using 59 paired adjacent non-cancerous tissue samples of NSCLC and serum samples of LC patients and healthy people and found that circSATB2 promotes the progression of NSCLC cells. They established the possible potential of circSATB2 as a biomarker of metastasis and defined the role of this circRNA in the proliferation, migration, and invasion of NSCLC. The study revealed that the levels of seven different circRNAs in both normal human bronchial epithelial cell lines (16HBE and BEAS-2B) and NSCLC cell lines (H226, H1299, H460, MES-1, A549, and H661) were measured. It was observed that a particular circRNA known as circ0008928, or circSATB2, exhibited a consistent and elevated presence in the NSCLC cell lines H1299, A549, and H460 when compared to the BEAS-2B normal cells. This suggests that circSATB2 may play a significant role in the biology of NSCLC. circSATB2 is located on chromosome 2q33.1 and consists of exons 4–8 of *SATB2*. It is involved in intercellular communication through exosomes. It is known that circSATB2 can regulate the expression of fascin homolog 1 and actin-binding protein 1 (FSCN1) through direct binding to miR-326, which additionally affects LC progression. Another group of researchers found that hsa_circ_0014130 is highly expressed in tissues during NSCLC and functions as an oncogene and leads to tumour growth by upregulating Bcl-2, in part via “sponging” miR-136-5p during NSCLC [181].

Researchers from Harbin have found that circRNA-002178, which can be detected in the plasma exosomes of LUAD patients, may serve as a promising non-invasive marker for early diagnosis of the disease. The research involved evaluating circRNA expression patterns in LUAD samples and their non-malignant adjacent counterparts. Out of these, three circRNAs were notably elevated. In that group, hsa-circRNA-002178 showed a consistent increase not only in LUAD samples but also across three LUAD cell lines: 95D, PC9, and A549. Data from TCGA revealed potential interaction sites between miR-133a-3p, miR-30c-3p, miR-34a, and circRNA-002178. Further investigations utilising qRT-PCR on 20 LUAD tissue pairs revealed a significant reduction in miR-34a in LUAD-affected tissues. The study’s findings indicate that circRNA-002178 can bind with miR-34a within tumour cells, potentially augmenting PD-L1 expression by acting as an miR-34 sponge in the cancer cells. Moreover, this group of researchers showed the possibility of circRNA-002178 delivery into T cells to stimulate PD1 expression by sequestering miR-28-5p through exosomes secreted by cancer cells [186].

Another group of Chinese scientists found that circPVT1 acts as a novel diagnostic biomarker or target for the treatment of individuals diagnosed with lung SCC and proved that circPVT1 promotes lung SCC progression through the HuR/circPVT1/miR-30d and miR-30e/*CCNF* cascade, performing RNA expression profile analysis in eight lung SCC tissues using sequencing (RNA-seq) techniques. Moreover, it appeared that lung SCC patients with higher circPVT1 expression levels showed shorter survival [193].

Zong Q. et al. proved that circRNA_102231 is one of the most significantly activated circRNAs and that circRNA_102231 expression was significantly amplified in LUAD tissues and associated with an advanced TNM stage, metastasis to LNs, and poor OS among LC patients [194].

In 2016, microarrays were used for the first time to search for candidate tumour-specific circRNA markers in LUAD tissues. Fresh tumour samples of primary LUAD and non-cancerous tissues were collected for analysis, and then the identified candidates were validated using 49 LUAD samples isolated from paraffin blocks. The authors found 39 circRNAs with increased expression and 20 miRNAs with a reduced level of expression. Moreover, it was found that hsa_circ_0013958 was highly expressed in all samples of LUAD—both in tissues and in plasma. Moreover, expression levels were correlated with the TNM stage (*p* = 0.009) and lymphatic metastasis (*p* = 0.006), with the area under the receiver operating characteristic curve being 0.815 (95% CI = 0.727–0.903; *p* < 0.001).

hsa_circ_0013958 has been shown to promote cell migration and invasion into LUAD cells. This was proven by disabling hsa_circ_0013958 in A549 and H1299 where migration and invasion inhibition was observed. Thus, the researchers concluded that hsa_circ_0013958 can be used as a potential non-invasive biomarker for the early detection and screening of LUAD [178].

### 4.2. circRNAs as Prognostic Biomarkers

The vast majority of prognostic marker investigations in LC are based on the analysis of circRNA expression in tissues. A significant number of studies have been conducted on LUAD [195,196,197,198,199].

Thus, it has been shown that in this histological type of LC, the expression of hsa_circ_0002346 is significantly reduced compared to normal lung tissues. Further studies using LUAD cell lines A549, H1299, H2228, pc9, and H1975 performed by Lin Wang’s group proved that circCRIM1 suppresses the invasion and metastasis of LUAD. The research encompassed multiple techniques and strategies carried out in laboratory conditions with live organisms, demonstrating that circCRIM1 can enhance the production of the recognised cancer-inhibiting agent, leukemia inhibitory factor receptor, by absorbing miR-182 and miR-93. Clinical and pathological studies indicated that lower levels of circCRIM1 in LC were closely associated with the spread of cancer to LNs and the TNM classification and were independently predictive of patients’ OS [197].

A different study indicated that the levels of circPVT1 were elevated in LUAD, and its presence was linked with advancement in the N stage and resistance to chemotherapy (specifically pemetrexed and cisplatin) among LC patients. Consequently, circPVT1 could potentially act as a predictive indicator for individuals afflicted with LUAD. Patients categorised into groups based on low and high circPVT1 expression demonstrated five-year OS rates of 53.57% and 33.33%, respectively. Furthermore, the Kaplan–Meier method revealed that patients with LC showing higher levels of circPVT1 had a reduced lifespan in comparison with those exhibiting lower levels. The use of Cox regression analysis verified that circPVT1 levels independently predicted outcomes for patients with LC. Additionally, elevated circPVT1 expression was identified in the A549/DR LC cell line that was resistant to chemotherapy, including cisplatin and pemetrexed. Diminishing circPVT1 in these cells increased their susceptibility to these chemotherapy drugs. In these cells, circPVT1 competed as an endogenous RNA against miR-145-5p. This particular miRNA was found to be underexpressed in LC tissues and cell lines resistant to cisplatin and pemetrexed. Through luciferase reporter assays, *ABCC1* was validated as the direct target of miR-145-5p in the A549/DR cells. Hence, the miR-145-5p/*ABCC1* axis is implicated in mediating drug resistance prompted by circPVT1 suppression in LUAD cells, marking circPVT1 as a conceivable marker for disease treatment prognosis [199].

A recent study concerning the expression of the circRNA CDR1 antisense RNA (*CDR1-AS*) gene, located at Xq27.1, CiRS-7, showed its upregulation in LUAD tissues and cell lines, which correlated with smoking history, T stage, and neoadjuvant chemotherapy in patients with LUAD. *CDR1-AS* was upregulated in PTX- and CDDP-resistant LC tissues and cell line A549/CR. Silence of *CDR1-AS* re-sensitised A549/CR cells to neoadjuvant chemotherapy. *CDR1-AS* plays a role in the EGFR/PI3K signaling pathway in A549/CR cells and might be an independent prognostic biomarker for LUAD patients [195].

One of study showed that circTUBGCP3 promotes LC progression. Increases in the circTUBGCP3 expression level or decreases in miR-885-3p expression were associated with pathological staging and poor survival in LC. Restoration of circTUBGCP3 expression resulted in the growth and invasion promotion of LUAD cells. circTUBGCP3 can act as a “sponge” for miR-885-3p, the role of which is to suppress cell proliferation and weaken the tumour-promoting effect of circTUBGCP3. Wnt10b, a target of miR-885-3p, may be upregulated by circTUBGCP3 and indicate poor survival in LC patients and points to the poor survival of patients with LC [198].

To assess the involvement of circRNAs in reactions to chemotherapy in SCLC, Weimei Huang and colleagues (2020) conducted a study analyzing gene expression through microarrays. They discovered that the circRNA known as cESRP1 (circRNA epithelial splicing regulatory protein1) may act as an indicator for predicting treatment outcomes and could be a viable therapeutic target for SCLC patients. Their research revealed a marked decrease in cESRP1 levels in cells that were resistant to chemotherapy compared to those that were sensitive to the treatment. cESRP1 appears to increase the efficacy of drugs by suppressing the activity of miR-93-5p in SCLC cells. In the cytoplasm, cESRP1 can directly interact with miR-93-5p, preventing its ability to repress gene expression post-transcriptionally. It does this by boosting the production of miR-93-5p target genes such as Smad7/p21(*CDKN1A*) and by interfering with the TGF-β-mediated EMT. These findings suggest that cESRP1 could be a valuable predictor of treatment response and a possible target for therapy in individuals afflicted with SCLC.

Regarding NSCLC, the biomarker circ10720 has been identified as a predictor of relapse in early-stage patients who have not received treatment. Using a custom TaqMan assay, the expression and regulation of circ10720 were investigated in four NSCLC cell lines (HCC44, A549, H23, and H1299) and a normal immortalised lung cell line (BEAS2B). The results revealed that circ10720 was overexpressed in the NSCLC cell lines HCC44 and A549 compared to BEAS2B, while it was downregulated in H23 and H1299 cell lines. Additionally, the impact of circ10720 on the EMT, apoptosis, and proliferation was assessed. Wound-healing and cell invasion assays demonstrated that the downregulation of circ10720 was associated with reduced migration and invasion capacities in HCC44 (*p* = 0.037 and *p* = 0.0035, respectively) and A549 (*p* = 0.0169 and *p* = 0.0470, respectively). Furthermore, an analysis of a cohort of 119 NSCLC patients who underwent surgical resection revealed that circ10720 was upregulated in tumor tissue compared to normal tissue (*p* < 0.001). The expression of circ10720 also varied among different histological subtypes (ANOVA *p* = 0.059), with higher expression observed in SCC compared to LUAD (*p* = 0.044). Moreover, circ10720 expression was found to be higher in current and former smokers than in people who had never smoked (*p* = 0.046). Patients with *TP53* mutations displayed higher levels of circ10720 compared to those with wild-type *TP53* (*p* = 0.0099). Notably, higher levels of circ10720 were associated with LN involvement (N+) (*p* = 0.0429) and a higher relapse rate [196].

### 4.3. circRNAs as Therapeutic Biomarkers

Expression of circFGFR1 is linked to negative clinical and pathological features and outcomes in patients with NSCLC. The circRNA known as circFGFR1 originates from the gene-encoding fibroblast growth factor receptor 1, which is implicated in tumour progression. Elevated levels of circFGFR1 enhance the capabilities of NSCLC cells to migrate, invade, multiply, and avoid immune detection, leading to unfavorable prognosis for patients with NSCLC. Therefore, inhibiting the circFGFR1-related pathways can have some potential in therapy for NSCLC [200]. circRNAs can promote the TGF-β-induced EMT and invasion in NSCLC. It has been shown that circRNA produced from the *PTK2* gene hsa_circ_0008305, or circPTK2, controls transcriptional intermediary factor 1 γ (TIF1γ) in NSCLC, thus suppressing the TGF-β-induced EMT and tumour metastasis. Therefore, circPTK2 overexpression could provide a therapeutic strategy for advanced NSCLC [201].

Recently, it was revealed that fusion genes can also produce circRNAs involved in tumour development. One example is F-circEA generated from the *EML4-ALK* fusion gene. A study by Tan and colleagues demonstrated that its tumour promoting function is realised through the activation of cell migration and invasion [202]. Moreover, they showed its expression in cytoplasm, meaning F-circEA could serve as a marker in liquid biopsies to control the *EML4-ALK* fusion gene in NSCLC [202]. Another circRNA derived from *EML4-ALK* fusion gene is F-circEA-2a. It also enhances cell migration and invasion but has little effect on cell proliferation. Moreover, it is found in tumour tissues but not in the blood of NSCLC patients [203]. circRNA hsa_circ_0000190 (C190) with oncogenic properties is also involved in the development of LC. It is directly involved in EGFR–MAPK–ERK signaling. C190 levels were upregulated in the blood samples of patients with advanced-stage LC, and its persistent expression in blood may predict a poor outcome of anti-PD-L1 treatment in patients. Thus, C190 may serve as a potential therapeutic target for the treatment of NSCLC [204].

### 4.4. circRNAs in Drug Resistance and Metastasis

The role of metabolic reprogramming in tumours is gradually gaining more attention and being investigated. Recent studies have shown that ncRNAs are strongly related to metabolic reprogramming. ncRNAs are able to directly regulate the expression and function of metabolic enzymes or indirectly regulate them through a number of important pathways regulating metabolism in cancer cells. It is possible that the study of key mechanisms that contribute to the reprogramming of cancer cells will shed light on new pathways and key links in the occurrence of LC and will contribute to the development of new therapeutic methods.

circRNAs are recognised for their crucial roles in the progression of NSCLC. They demonstrate dual roles and can act as oncogenes and tumour suppressors depending on the type and potential stage of cancer. circRNAs modulate the EMT, proliferation, cell cycle progression, invasion, and metastasis in different ways by sponging miRNAs and binding with RBPs. In the same ways, circRNAs modulate drug resistance.

Chemotherapy represents a widely employed approach for cancer treatment, encompassing NSCLC. It has been observed that cells exhibiting resistance to specific chemotherapeutic agents may also display resistance to other drugs with distinct structures through diverse mechanisms, demonstrating multidrug resistance. While some factors within tumour cells contributing to the development of resistance to chemoradiation and targeted therapy have been identified, the comprehensive understanding of the process and the underlying molecular mechanisms remains incomplete. Recent investigations have delineated the involvement of circRNAs in developing of drug resistance in NSCLC. In the study of Dexun Hao et al., 2023, the impact of circ_0110498 on cisplatin (DDP) resistance in NSCLC was investigated. The analysis included the measurement of glucose consumption and lactate production to assess cell glycolysis. Western blot analysis determined protein expression levels. The expression of circ_0110498, miR-1287-5p, and *RBBP4* was assessed. They observed upregulation of circ_0110498 in DDP-resistant NSCLC tissues and cells. Silencing circ_0110498 not only suppressed DDP resistance in NSCLC cells by inhibiting cell growth, metastasis, and glycolysis but also enhanced DDP sensitivity in NSCLC tumours. circ_0110498 was found to sponge miR-1287-5p, and the inhibitor of miR-1287-5p reversed the effect of circ_0110498 silencing on DDP resistance in NSCLC cells. miR-1287-5p was identified to interact with *RBBP4*, and overexpression of *RBBP4* partially reversed the inhibitory effect of miR-1287-5p on DDP resistance in NSCLC cells. Their findings suggest that circ_0110498 contributes to DDP resistance in NSCLC, at least in part through mediating the miR-1287-5p/*RBBP4* signaling pathway [205].

One study focused on circPTK2, miR-942, and tripartite motif 16 (TRIM16) in the context of NSCLC resistance. In NSCLC tissues and cell lines, circPTK2 (hsa_circ_0008305) and *TRIM16* exhibited low expression, while miR-942 showed significant upregulation. Overexpression of circPTK2 notably inhibited cell growth, metastasis, and glycolysis in A549/CDDP and H1299/CDDP cells. The effects of high circPTK2 expression on cell growth, metastasis, and glycolysis were reversed by promoting miR-942 or inhibiting *TRIM16* in A549/CDDP and H1299/CDDP cells. In vivo experiments demonstrated that circPTK2 overexpression inhibited the growth of A549/CDDP cells. Furthermore, circPTK2 was found to attenuate cisplatin (CDDP) resistance in NSCLC by modulating the miR-942/*TRIM16* axis. This provides a novel perspective for NSCLC treatment and enhances our understanding of the mechanism underlying CDDP resistance in NSCLC [206]. To investigate the role of circRNA circHIPK3 in conferring resistance to gefitinib in LC cells, Yi Zhao et al., 2022, enrolled 110 LC patients for the study. Tissue samples from each subject, both cancerous and nearby non-tumorous tissue, were collected and embedded in paraffin to examine the levels of circHIPK3 in various tissues. To study how cells respond to different conditions in terms of apoptosis, an LC cell line resistant to gefitinib was created. Findings showed that the circHIPK3 expression was notably reduced in patients having tumours 3 cm or larger compared to those with smaller tumours (*p* < 0.05). Moreover, the circHIPK3 expression levels were considerably lower in patients at TNM stage II/III than in those at stage I (*p* < 0.05). For patients with LN metastasis, circHIPK3 levels were significantly decreased compared to those without metastasis (*p* < 0.05). Out of the different TNM stages of LC tissues analyzed, only six patients demonstrated high circHIPK3 expression, while low levels were observed in the remaining 104 patients. The apoptosis rate of gefitinib-mediated LC drug-resistant cell lines notably decreased. Thus, circHIPK3, a circRNA, demonstrated significantly reduced expression in LC tissues. Its low expression appears to enhance drug resistance in LUAD cells to gefitinib [207]. It was also shown that circRNA_103762 is upregulated in cisplatin-resistant H358/CDDP LC cells. It represses the expression of DNA damage inducible transcript 3 and facilitates multidrug resistance in NSCLC [208]. The elevation of hsa_circ_0001946 levels enhances the responsiveness of A549 cells to cisplatin. Additionally, the inhibition of hsa_circ_0001946 leads to the activation of the NER signaling pathway, resulting in reduced cisplatin sensitivity in LC. Hsa_circ_0001946 is intricately involved in modulating the responsiveness of NSCLC cells to cisplatin through its influence on the NER signaling pathway. Hsa_circ_0001946 acts as a sponge for four miRNAs (hsa-miR-7-5p, hsa-miR-671-5p, hsa-miR-1270, and hsa-miR-3156-5p) to finely regulate the NER signaling pathway [209]. circ_0002483 exhibits decreased expression levels in tissue samples of NSCLC as well as in Taxol-resistant NSCLC cell lines. Diminished levels of circ_0002483 correlate with an unfavorable prognosis for NSCLC patients. The Cell Counting Kit-8 assay revealed that the increased level of circ_0002483 significantly boosted the sensitivity of NSCLC cells to the chemotherapy drug Taxol. Using an RNA immunoprecipitation assay and dual-luciferase reporter assays, the research confirmed that circ_0002483 binds to miR-182-5p in a competitive manner. Diminishing miR-182-5p levels enhanced the susceptibility of A549 and H1299 cell lines to Taxol. Further investigation using a luciferase assay showed that miR-182-5p has the capability to attach to the 3′ untranslated regions of several proteins, including forkhead box O3 (FOXO3), forkhead box O1 (FOXO1), and growth factor receptor-bound protein 2 (GRB2). When miR-182-5p was co-transferred with circ_0002483 into cells, the protein expressions of FOXO1, FOXO3, and GRB2 were reestablished, which in turn led to a heightened resistance to Taxol in NSCLC cells. Hence, these findings indicate that circ_0002483 serves as a “sponge” for miR-182-5p, which ultimately mediates the upregulation of GRB2, FOXO1, and FOXO3, augmenting the reaction of A549 and H1299 cells to Taxol [210]. The circRNAs hsa_circ_0092857 and hsa_circ_0004350 are significantly linked to the control of protein synthesis. Analysis through gene ontology has shown that the overlapping RBPs related to both circEIF3as act as translation regulators and might interact in a complementary way with their parental gene, EIF3a. An imbalance in the levels of hsa_circ_0092857 and hsa_circ_0004350 could potentially alter the response to cisplatin in LC cells [211]. Tumour metastasis is a complex procedure where cancerous cells spread beyond their original location. These cells penetrate the basement membrane and move from their origin to new locations via the lymphatic and blood systems or through body cavities, usually homing in on distinct organs or structures. As a result, they form a new tumour at a different site, which typically bears similar histological characteristics as the original one. The EMT process is crucial and highly correlated with the pathological changes seen in tumour expansion [212]. Particularly, the EMT is considered the main instigator behind the invasive and metastatic behaviour of tumours. The spread of LC cell metastasis is a serious obstacle that reduces OS in LC patients and marks a key stage in LC’s malignant development. Moreover, the metastasis to LNs is a primary pathway for the dispersal of NSCLC and is greatly indicative of a poor prognosis in those afflicted. The regulatory mechanisms governing lymphangiogenesis and metastasis in NSCLC remain unclear. Xiayao Diao et al., 2023, identified the circRNA circTLCD4-RWDD3, which exhibited significant upregulation in extracellular vesicles (EVs) derived from LN metastatic NSCLC. This circRNA was positively correlated with decreased OS and disease-free survival (DFS) in a multicenter clinical cohort of NSCLC patients. Silencing the expression of EV-packaged circTLCD4-RWDD3 demonstrated inhibitory effects on both in vitro and in vivo lymphangiogenesis and LN metastasis in NSCLC. At the molecular level, circTLCD4-RWDD3 was found to transcriptionally upregulate the expression of ubiquitin carrier protein 9 (UBC9), facilitating SUMO2 modification on the K108 residue of hnRNPA2B1. This modified hnRNPA2B1 was subsequently recognised by the SUMO interaction motif (SIM) sequence within ALG-2-interacting protein X (ALIX), activating ALIX. This activation led to the packaging of circTLCD4-RWDD3 into EVs by recruiting endosomal sorting complexes required for transport (ESCRT)-III. The EV-packaged circTLCD4-RWDD3 was then transmitted to HLECs, where it activated the transcription of prospero homeobox 1 (PROX1), inducing lymphangiogenesis and LN metastasis. These findings unveil a novel mechanism involving the loading of circTLCD4-RWDD3 into EVs derived from NSCLC cells, promoting lymphangiogenesis. The study suggests that EV-packaged circTLCD4-RWDD3 may serve as a potential therapeutic target for addressing lymphatic metastasis in NSCLC [213]. The study revealed elevated levels of circ_0000043 in 16HBE-T cells, which are human bronchial epithelial cells turned malignant due to benzo[a]pyrene-trans-7,8-diol-9,10-epoxide exposure. This overexpression was later verified to be significantly higher in LC cell lines and affected lung tissues. The research also indicated a direct relationship between high levels of hsa_circ_0000043 and negative clinical and pathological features, such as an advanced tumor-node metastasis stage, LN involvement, distant metastasis, and the OS rate. Laboratory tests showed that reducing hsa_circ_0000043 levels hindered the growth, invasive potential, and movement of the 16HBE-T cells. These findings were consistently observed in a mouse model where tumor growth was suppressed. The investigations further revealed that hsa_circ_0000043 captures miR-4492, resulting in its decreased expression, which is connected to unfavorable clinical and pathological outcomes. The study concluded that hsa_circ_0000043 escalates the growth, transformation to malignancy, mobility, and invasive behavior of 16HBE-T cells by trapping miR-4492, suggesting that brain-derived neurotrophic factor (BDNF) and the activation of signal transducer and activator of transcription 3 (STAT3) are fundamental in these mechanisms [214]. LUAD exhibits a notable propensity for bone metastasis. In one of the studies, bone metastasis-associated circ_0096442 was identified with the use of bioinformatic analyses. Overexpression of circ_0096442 was found to reduce apoptosis in A549 cells compared to control groups, while reducing its expression had the opposite effect. Elevated levels of circ_0096442 also markedly decreased the protein levels of those that encourage apoptosis and increased levels of *BCL2*, which is an anti-apoptosis gene. Additionally, enhanced circ_0096442 expression notably improved the cells’ capacity to heal wounds, a capability that was substantially reduced when circ_0096442 was knocked down. Thus, the overexpression of circ_0096442 seems to aid the growth, invasiveness, and movement of A549 cells within the bone environment [215]. In the research of Xinyi Ma et al., 2023, they conducted miRNA deep-sequencing to identify differentially expressed serum miRNAs in NSCLC and to construct a circRNA–miRNA network specific to NSCLC for the assessment of its diagnostic potential. The circRNA–miRNA network, encompassing hsa-miR-4482-3p, hsa-miR-146a-3p, hsa_circ_0008167, and hsa_circ_0003317, was constructed based on their interactions and preliminary testing in NSCLC cells. Quantitative real-time polymerase chain reaction (qRT-PCR) was utilised to quantify the relative levels of the selected ncRNAs in healthy individuals, pneumonia patients, those with benign lung tumors, and NSCLC cohorts. ROC analysis was conducted to evaluate the diagnostic potential of the circRNA–miRNA network. The dysregulation of serum levels of hsa-miR-146a-3p, hsa-miR-4482-3p, hsa_circ_0003317, and hsa_circ_0008167 has been observed in individuals affected by NSCLC. These four ncRNAs together provided the greatest accuracy in differentiating NSCLC from benign lung growths. Furthermore, a high presence of hsa_circ_0008167 was associated with more advanced and aggressive forms of NSCLC, marked by the spread to LNs, distant organs, and an advanced stage of cancer. Moreover, the integration of hsa-miR-4482-3p with hsa_circ_0008167 increased the ability to discern between the presence and absence of LN metastases, while the combination of hsa-miR-4482-3p with both hsa_circ_0008167 and hsa-miR-146a-3p was more effective at identifying the stage of the disease. These findings outperformed the diagnostic capabilities of individual circRNAs or miRNAs, as well as traditional cancer indicators. Thus, a specific network of circRNA–miRNA was discovered, which can be highly effective and accurate for the diagnosis of NSCLC, offering a pioneering approach for the creation of diagnostic markers related to competing endogenous RNAs in a range of cancers [216].

## 5. Conclusions

This review presents the diversity of ncRNAs; their biogenesis, function, and possible targets are described, as well as their role in the regulation of cellular processes. Undoubtedly, all these mechanisms represent complex and, in many ways, poorly understood processes, in which there are still quite a lot of unknowns. However, a great deal of effort is being put into understanding the mechanisms of these RNAs.

Thus, studies in recent years have led to a broadening of the spectrum of ncRNAs, and it has become clear that the most important roles of ncRNAs in the cell are epigenetic control and the regulation of protein-coding genes (Figure 5). Although the functions of many ncRNAs have yet to be fully characterised, many have shown specific expression, localization to specific regions, and association with cancer in humans. It is becoming increasingly clear that ncRNAs can function in a variety of ways and are key regulatory molecules in cells. For example, lncRNAs can produce small RNAs or modulate the processing of other RNAs. We owe much of the rapid progress in the discovery of new classes of ncRNAs and the accumulation of knowledge about ncRNAs to the capabilities of high-throughput sequencing and big data technologies, which will continue to be essential tools for solving medical genetics problems in the future. Despite the annually increasing number of studies on the role of ncRNAs in various types of cancer, their use as biomarkers in the clinic is not possible in the near future. There are several limitations for that nowadays: the absence of markers with high specificity and sensitivity, conflicting research results, a large number of ncRNA targets, the inability to regulate the expression of many genes, and the limited sample sizes used for research. However, in the future, by combining several types of markers, taking into account the clinical and pathological parameters of the patient and the tumour, ncRNAs may be useful for the diagnosis, prognosis of the course of the disease, and choice of therapy in patients with LC.

The accumulated data leave no doubt that these molecules have prospects for practical application. Altered regulation of ncRNAs can be detected in body fluids of patients, and their distinctive expression patterns are strongly associated with certain pathological features, tumor-free survival, OS, and DFS. This suggests that ncRNAs hold promise as noninvasive biomarkers and potential therapeutic targets in the treatment of LC. Nonetheless, prior to their application in clinical interventions, further research should concentrate on elucidating the precise functions of ncRNAs in LC pathogenesis and devising novel therapeutic strategies based on ncRNAs.

## Figures and Tables

**Figure 1 ijms-25-00560-f001:**
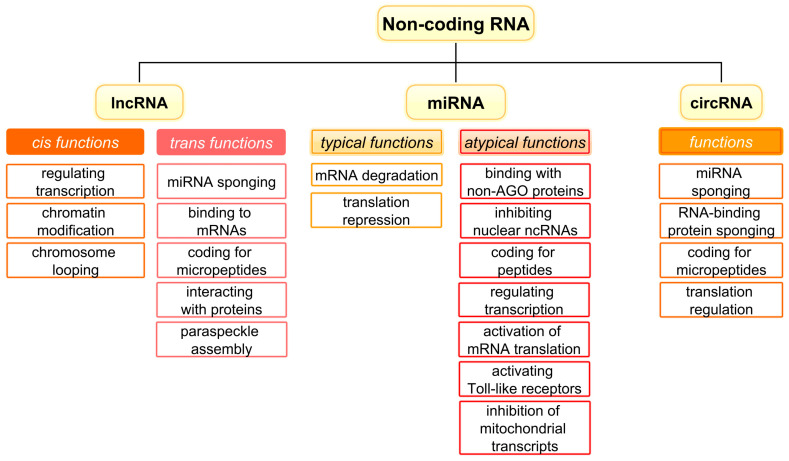
Functions of the main ncRNA classes. mRNA—messenger RNA; AGO proteins—Argonaute proteins.

**Figure 2 ijms-25-00560-f002:**
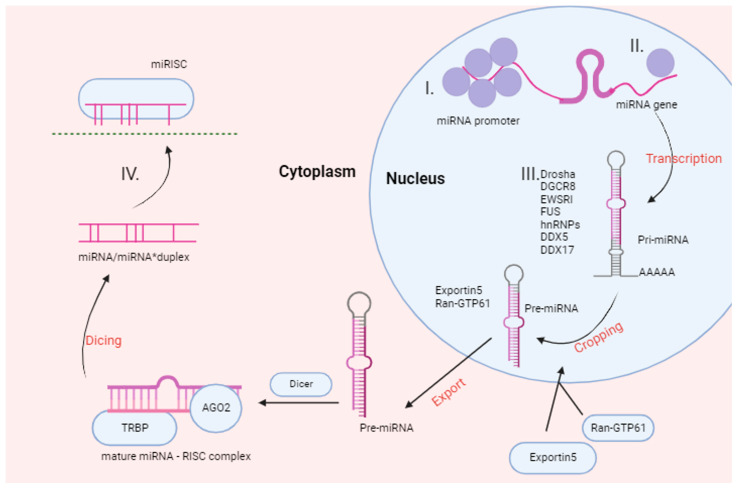
miRNA biogenesis. hnRNPs—heterogeneous nuclear ribonucleoproteins; miRISC—miRNA-induced silencing complex; TRBP—transactivation response element RNA-binding protein. **I**–**II**—synthesis of the primary miRNA (pri-miRNA) by polymerase II/III; **III**—processing of pri-miRNA into miRNA precursor (pre-miRNA); **IV**—the formation of RISC complex.

**Figure 3 ijms-25-00560-f003:**
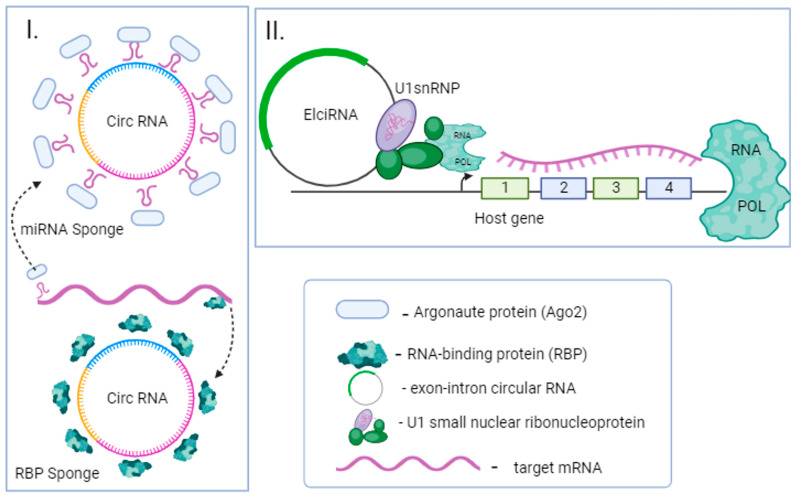
Mechanisms of circRNA functioning. (**I**) One prominent function of circRNAs is their ability to act as molecular sponges for miRNA/RBP, effectively sequestering these molecules away from their mRNA targets. This leads to gene expression alterations. (**II**) circRNAs can interact with U1 snRNP, facilitating their interaction with transcription complexes at host genes and subsequently inducing transcription of these genes.

**Figure 4 ijms-25-00560-f004:**
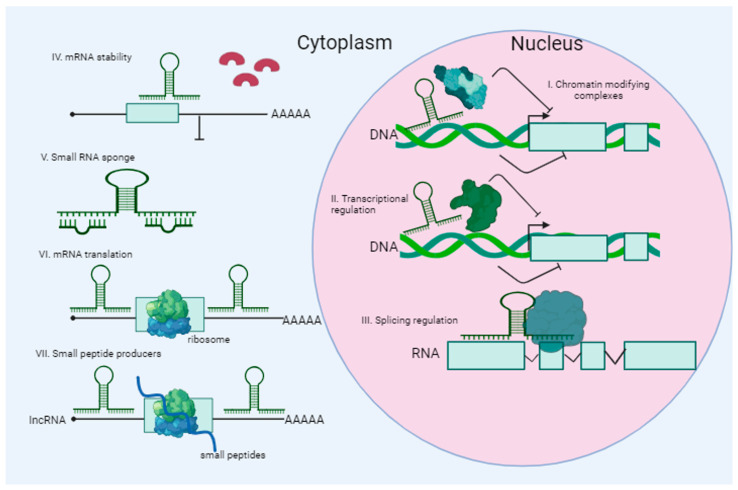
Functions of lncRNAs. The essential roles of lncRNAs encompass various biological processes. In the nuclear compartment, lncRNAs are implicated in the regulation of epigenetic mechanisms, transcriptional processes, and splicing operations. Conversely, in the cytoplasmic domain, lncRNAs are involved in maintaining mRNA stability, acting as modulators for small regulatory RNAs, regulating mRNA translation, and exhibiting the potential to generate small peptides.

**Figure 5 ijms-25-00560-f005:**
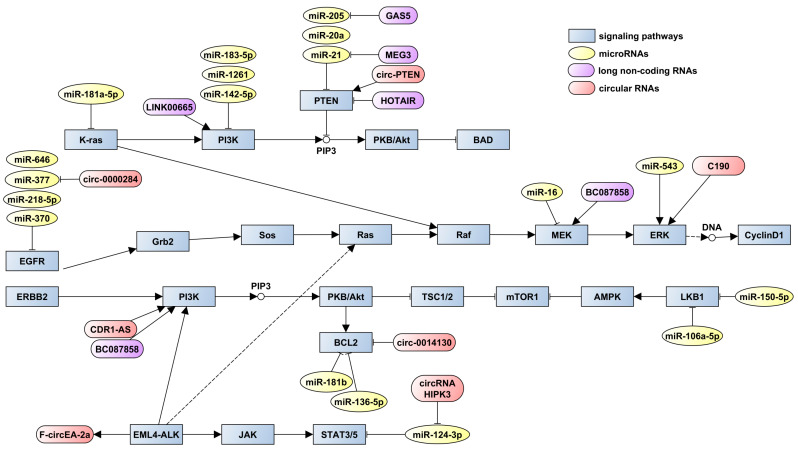
Schematic map of key NSCLC signaling pathways influenced by ncRNA regulators. Abbreviations: K-ras—protein encoded by the KRAS (Kirsten rat sarcoma viral) gene; PI3K—phosphatidylinositol 3-kinase; PIP3—phosphatidylinositol-3,4,5-triphosphate; PTEN—phosphatase and tensin homolog; PKB/Akt—protein kinase B, also known as Akt; BAD—Bcl-2-antagonist of cell death; EGFR—epidermal growth factor receptor; Grb2—growth factor receptor-bound protein 2; Sos—Son of Sevenless; TSC1/2—tuberous sclerosis complex 1/2; mTOR—mammalian target of rapamycin; AMPK—AMP-activated protein kinase; LKB1—liver kinase B1; EML4—echinoderm microtubule-associated protein-like 4; JAK—Janus kinase; STAT—signal transducer and activator of transcription.

**Table 1 ijms-25-00560-t001:** Functional effects and clinical significance of lncRNAs.

Effects/Significance	LncRNAs	References
Tumour promoters	HOTAIR	[124,125,126]
	MALAT1	[127]
	PVT1	[128]
	LUCAT1	[129,130]
	LINC00680	[131,132]
	LINC00511	[133]
Tumour suppressors	GAS5	[134]
	TUG1	[135]
	MEG3	[136]
Diagnostic and prognostic markers	UCA1	[137]
	MALAT1	[138,139]
	HOTAIR	[126]
	LUCAT1	[129]
	SPRY4-IT1	[140]
	NEAT1	[141]
Therapeutic targets	PVT1	[128]
	UCA1	[137]
	H19	[142]
	LINC00665	[143]
	BC087858	[144]
	MEG3	[145]

**Table 2 ijms-25-00560-t002:** Role of circRNAs in tumour biology.

Process	circRNAs	References
EMT metastasis	Circ-0043265/miR-25-3p/*FOXP2*	[169]
	circ-PTPRA/miR-96-5p	[170]
	RASSF8/E-cadherin	[171]
Proliferation	circ-FOXM1/miR-614/*FAM83D*	[172]
	circ-ARHGAP10/miR-150-5p/*GLUT1*	[173]
	circ-0000326/miR-338-3p/*RAB14*	[174]
	circ-0102231/miR-145/*RBBP4*	[175]
	circ-ABCB10/miR-584-5p/*E2F5*	[176]
Cell cycle	circ-Foxo3-p21-CDK2-*CDK2*	[177]
	circ-0013958/miR-134/cyclin D1	[178]
	circ-0006916/miR-522	[179]
	circ-TP63/miR-873-3p/*FOXM1*	[180]
Apoptosis	circ-0014130/miR-136-5p/*BCL2*	[181]
	circ-0074027/miR-185-3p/*BRD4*	[182]
	circ-0014130/miR-142-5p/IGF-1	[183]
	circ-0043265/miR-25-3p/*FOXP2*	[169]
Autophagy	circular RNA HIPK3/miR-124-3p/*STAT3*	[184]
	circ-0085131/miR-654-5p/*ATG7*	[185]
Microenvironment	circ-002178/miR-34/PDL1	[186]
	circ-CPA4/let-7/miRNA/PD-L1	[187]
	circ-0000284/miR-377/PD-L1	[188]
	circ-PIP5K1A/miR-600/HIF-1a	[189]
	circ-SLC25A16/miR-488-3p/HIF-1a	[190]

## Data Availability

No new data were created or analyzed in this study. Data sharing is not applicable to this article.

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
