# Peer review of "Non-Coding RNAs as Key Regulators in Lung Cancer"

_ijms, 2023, doi:10.3390/ijms25010560_

Round 1
Reviewer 1 Report
Comments and Suggestions for Authors
This paper offers an insightful review shedding light on the pivotal role of non-coding RNAs (ncRNAs) as regulators in lung cancer. The comprehensive overview encompasses recent studies delving into the multifaceted functions, regulatory mechanisms, and therapeutic potential of various ncRNAs, such as microRNAs (miRNAs), long non-coding RNAs (lncRNAs), and circular RNAs (circRNAs) across different types of lung cancer.
However, some areas could benefit from further clarity and detail. In the introduction, expanding on the biogenesis and distinct characteristics of each type of ncRNA would provide readers with a clearer understanding. Incorporating a summarizing figure could aid in illustrating these differences effectively.
Figures 1 to 3 could be improved to provide clearer information. The representation of all connected target points lacks clarity, and replacing these with tables and including references for each target might enhance their effectiveness. Additionally, an overall schematic depicting the signaling pathways involved in lung cancer, influenced by ncRNA regulators, would provide readers with a comprehensive overview quickly.
Addressing these suggestions could significantly enhance the quality and comprehensibility of this review paper.
Author Response
Dear Reviewer1,
Thank you for a careful analysis of our article and valuable comments. The article has been revised according to the recommendations and comments:
- In the introduction we described the biogenesis of non-coding RNAs and their distinct characteristics and in addition we have created several pictures demonstrating their roles and biogenesis to give readers a clearer understanding.
- We removed Figures 1 to 3 and made a new overall schematic figure depicting the signaling pathways involved in lung cancer, influenced by ncRNA regulators to provide readers with a comprehensive overview
Thank you for your great efforts to improve the quality of our paper.
Reviewer 2 Report
Comments and Suggestions for Authors
The article titled "Non-coding RNAs as Key Regulators in Lung Cancer" provides an insightful exploration into the intricate role of non-coding RNAs (ncRNAs) in the context of lung cancer. The authors delve into the diverse landscape of ncRNAs, including microRNAs (miRNAs), long non-coding RNAs (lncRNAs), and circular RNAs (circRNAs), shedding light on their potential as crucial regulators in the pathogenesis and progression of lung cancer. As I embark on the review of this article, I aim to offer constructive feedback and suggestions to enhance its clarity, comprehensiveness, and overall impact in the field.
I have several suggestions to enhance the quality of this article. Firstly, in the introduction, it would be beneficial to emphasize the clinical significance of the discussed ncRNAs and their potential applications in lung cancer diagnosis, prognosis, and treatment. Providing a brief overview of the clinical relevance in this section will engage readers and highlight the practical implications of the research.
Secondly, within the miRNA discussion, it is crucial to address their involvement in inducing resistance to targeted therapies and immunotherapy. This aspect is pivotal in the context of lung cancer treatment, and its inclusion will contribute to a more comprehensive understanding of the subject matter.
Thirdly, the section on lncRNAs could benefit from a more detailed elucidation of the molecular mechanisms underlying their functionality. Expanding on these mechanisms will not only enhance the readers' understanding but also add depth to the discussion of lncRNAs in lung cancer.
Fourthly, in the discussion of circRNAs, it is important to highlight their role in the metabolism and metastasis of cancer cells. Incorporating this information will provide a more holistic view of the impact of circRNAs on lung cancer biology.
Fifthly, he figures presented seem unsuitable for the main text and would be better suited for supplementary information. Instead of these current figures, I propose adding a single schematic diagram depicting a key cancer signaling pathway implicated in lung cancer, highlighting how various ncRNA classes influence and regulate components of this pathway.
Lastly, in the conclusion, addressing the clinical application challenges of ncRNAs in lung cancer and proposing potential solutions will add a forward-looking perspective. This inclusion will make the conclusion more impactful and open avenues for future research in the field.
Comments on the Quality of English LanguageThe quality of the manuscript is commendable.
Author Response
Dear Reviewer2,
Thank you for a careful analysis of our article and valuable comments. The article has been revised according to the recommendations and comments:
- In the introduction we have added information to emphasize the clinical significance of ncRNAs and their potential applications in lung cancer diagnosis, prognosis, and treatment.
- Within the miRNA discussion we have discussed their involvement in inducing resistance to targeted therapies and immunotherapy.
- In the section on lncRNAs elucidation of the molecular mechanisms underlying their functionality was added.
- We have added a new section devoted to the role of circRNAs in drug resistance and metastasis of cancer cells.
- We have created a single schematic
figure depicting a key cancer signaling pathway implicated in lung cancer, highlighting how various ncRNA classes influence and regulate components of this pathway. - In conclusion we have added clinical application challenges of ncRNAs in lung cancer.
Thank you for your great efforts to improve the quality of our paper.
Round 2
Reviewer 2 Report
Comments and Suggestions for Authors
It seems that the quality of the article has reached an acceptable level. Thank you for your effort.